# Validity and Effects of Placement of Velocity-Based Training Devices

**DOI:** 10.3390/sports9090123

**Published:** 2021-08-31

**Authors:** Raphael Fritschi, Jan Seiler, Micah Gross

**Affiliations:** 1Department of Medicine, Movement and Sport Science, University of Fribourg, 1700 Fribourg, Switzerland; raphaelfritschi92@gmail.com; 2Department for Elite Sport, Swiss Federal Institute of Sport Magglingen (SFISM), Hauptstrasse 247, 2532 Magglingen, Switzerland; jan.seiler@baspo.admin.ch

**Keywords:** velocity, barbell kinematics, accuracy, precision, inertial measurement unit (IMU), linear position transducer, laser, resistance training

## Abstract

Velocity-based training (VBT) is a resistance training method by which training variables are manipulated based on kinematic outcomes, e.g., barbell velocity. The better precision for monitoring and manipulating training variables ascribed to VBT assumes that velocity is measured and communicated correctly. This study assessed the validity of several mobile and one stationary VBT device for measuring mean and peak concentric barbell velocity over a range of velocities and exercises, including low- and high-velocity, ballistic and non-ballistic, and plyometric and non-plyometric movements, and to quantify the isolated effect of device attachment point on measurement validity. GymAware (*r* = 0.90–1, standard error of the estimate, SEE = 0.01–0.08 m/s) and Quantum (*r* = 0.88–1, SEE = 0.01–0.18 m/s) were most valid for mean and peak velocity, with Vmaxpro (*r* = 0.92–0.99, SEE = 0.02–0.13 m/s) close behind. Push (*r* = 0.69–0.96, SEE = 0.03–0.17 m/s) and Flex (*r* = 0.60–0.94, SEE = 0.02–0.19 m/s) showed poorer validity (especially for higher-velocity exercises), although typical errors for mean velocity in exercises other than hang power snatch were acceptable. Effects of device placement were detectable, yet likely small enough (SEE < 0.1 m/s) to be negligible in training settings.

## 1. Introduction

Velocity-based training (VBT) is a resistance training method by which training variables are manipulated based on kinematic outcomes, e.g., barbell velocity, of individual movement cycles (repetitions). In contrast to traditional methods using percentages of an assumed maximal tolerable resistance (commonly, a one-repetition maximum i.e., 1-RM) and predetermined numbers of repetitions, VBT facilitates gauging intensity by movement velocity [1] and set length by cumulative deterioration of velocity within the set [2]. The justification for VBT is that training variables can be determined or adjusted on an individual and day-to-day basis more precisely than with traditional methods [3]. This reasoning along with current advances in mobile technology are driving VBT’s quickly spreading acceptance among strength training coaches and athletes. Nonetheless, the claim of better precision for monitoring and manipulating training variables assumes that velocity can be measured and communicated correctly.

To assess the legitimacy of this assumption, numerous studies, recently summarized in the reviews of Weakley et al. [4] and Clemente et al. [5], have assessed the validity or reliability of the velocity parameters provided by various VBT mobile devices. Generally accepted are the parameters mean (***v***_mean_) and peak (***v***_peak_) concentric velocity (although mean propulsive velocity has also been proposed for fast, non-ballistic exercises [3]), which are usually obtained by linear position transducers (LPTs), inertial measurement units (IMUs), or some sort of optical system. Whereas Clemente et al. [5] point out that devices employing IMUs are often most practical to use, Weakley et al. [4] suggest that LPTs are generally more accurate, although a strict distinction does not exist. Despite the abundance of largely similar studies in this area, new devices continue to become commercially available while existing devices evolve from one version to the next. Moreover, the existing literature appears biased with its preference for simple, controlled, non-plyometric and often non-ballistic movements and may, therefore, overlook important practical challenges associated with measuring velocity in athletic training settings. Finally, a myriad of reference methods has been employed, whereas the use of a 3D motion capture systems as the gold-standard criterion measure in new validity studies has been highly recommended [4].

When assessing new or lesser-known devices, comparison with a more established mobile VBT device is helpful. The GymAware PowerTool (an LPT) has been assessed repeatedly and appears to be among the most valid [4,6,7,8], making this device a convenient means for comparison. The Push device (an IMU) has also been assessed often, generally being found to be less valid than GymAware [7,8]. In contrast, the laser-based Flex, which has been assessed in only one study and only for ***v***_mean_ [9], and the IMU-based Vmaxpro, which appears to be missing from the peer-reviewed literature altogether, lack impartial support for their validity. Another system, the 1080 Quantum, although not mobile, has also become common in athletic training settings but has been assessed in only one study, and not against a motion capture criterion measure [10].

In their review, Weakley et al. [4] recommend testing VBT devices over a wide range of velocities and exercises. In particular, these authors suggest including Olympic lifts with free weights, which has been the case in only a handful of studies [4,5]. In contrast to exercises such as bench press, squat, or deadlift, which have been employed by the majority of studies, Olympic lifts are characterized by a more complex bar path and velocity progression in the concentric phase, which could pose greater challenges to VBT devices. A further, seldom-considered factor is the effect of a stretch-shortening cycle immediately prior to the concentric phase, such as for countermovement jumps and other plyometric movements. Where plyometric movements have been included, validity appeared quite poor [11,12]. However, since plyometric exercises have been largely ignored, with the exception of a few studies on less common devices [11,12,13], little is known about how the rapid change in direction during exercises with a preceding stretch-shortening cycle affects concentric velocity measurement validity.

In addition to complex bar paths and rapid countermovements, a further challenge particular to measuring barbell velocity during free weight exercises could be non-parallel movement of the barbell or barbell bending [14]. Appleby et al. recently provided evidence that bar end displacement differs from that of the bar midpoint, as well as between right and left ends, thus suggesting that the point of reference along the barbell has implications for velocity measurements [14]. Some previous studies have ignored this issue, taking criterion measurements at the barbell’s end or midpoint and assuming parallel movement [8,15,16,17], while others have eliminated the issue by taking reference measures at the device attachment point [7,9,18]. However, no study has yet quantified the effect of non-parallel barbell motion itself on velocity measures (particularly, when these measures are made toward one end of the barbell). Although deviations from parallel movement are typically small (usually even imperceptible to the naked eye), the effects of these within the concentric phase would be expected to be greater for ***v***_peak_ than for ***v***_mean_, and to be more severe the further measurements are made from the barbell midpoint [14]. Due to the different technologies used by the various devices on the market, the attachment point of a device, and thus the location where measurements are made, may be near or at the end of the barbell. This could lead to over- or underestimation of bar velocity in practice, independent of a device’s ability to track its own movement correctly, if the bar is not kept parallel during the movement. To determine whether these effects are negligible or not, the effect of device placement must be assessed independently from a device’s technological precision and accuracy.

Thus, the aims of this study were to assess and compare the validity of several mobile VBT devices and one stationary VBT device for measuring mean (***v***_mean_) and peak (***v***_peak_) concentric barbell velocity over a range of velocities and free-weight exercises, including low- and high-velocity, ballistic and non-ballistic, and plyometric and non-plyometric movements, with reference to a three-dimensional motion-capture system. We aimed to assess the pure technological validity of each device, independent of its point of attachment to the barbell, and in doing so, also quantify the isolated effect of attachment point itself on measurement validity. We hypothesized that each device would provide valid velocity measurements for its own attachment point but that attachment points closer the barbell end would amplify discrepancies between device measures and barbell midpoint velocity.

## 2. Materials and Methods

All study procedures complied with the Declaration of Helsinki and were approved by the ethics committee of the Canton of Bern, Switzerland (project ID 2018-00742).

### 2.1. Participants and Study Design

Eleven men (28 ± 5 years old) and three women (24 ± 2 years old) were informed ahead of time of the aims, risks, and benefits of the study and volunteered while giving their written informed consent to participate. All participants were healthy and regularly physically active but represented a wide variety of free-weight training experiences, from beginner to professional instructor. The wide range of strength and technical competency within the participant cohort was intentional. In contrast to a cohort with abundant free-weight training experience, this variety was supposed to better reflect the ability levels represented by consumers of the tested devices.

Data were collected over the course of two days, whereby each participant attended only one test session. Upon arrival, participants were first informed in detail about the session, and then they proceeded to a warm-up. The warm-up was conducted mostly according to individual preferences; however, participants did a few repetitions of each tested exercise under the observation of (and if necessary, abiding by the corrections of) one of the investigators, to ensure they could execute each exercise with adequate technique. At the end of the warm-up, participants performed several repetitions of back squats, starting with a load they expected (based on experience) to be light or moderate, then with loads increasing by increments of 5–20 kg at a time, as suggested by an investigator, each with maximal voluntary concentric speed, while barbell velocity was monitored with a GymAware PowerTool (Kinetic Performance Technology, Mitchell, Australia) device. The goal here was to determine two individual loads corresponding to ***v***_mean_ in the range of 0.7–0.8 m/s and around 0.5 m/s, respectively, which would be used for squats during the main measurements to follow.

Following warm-up, participants proceeded to the main measurements. These were performed with a 2.20 m, 15 kg barbell (Eleiko, Halmstad, Sweden) equipped with several mobile VBT devices (see Table 1). Additionally, two motor-controlled cable-pull devices (1080 Quantum, 1080 Motion, Lidingö, Sweden) were attached to either end of the barbell, each providing a resistance of 2.5 kg (5 kg total). As such, the minimal additional load was 20 kg. Participants performed 1–2 sets of five repetitions each, separated by one minute of rest, for each of five exercise configurations. Participants were instructed to perform each repetition of each set with the maximal concentric velocity. To best reflect real-world usage of the VBT devices, eccentric velocity (−0.37 ± 0.14 m/s) was not standardized or manipulated. However, participants did reset to a stable, standing position between repetitions, rather than performing the repetitions within a set continuously.

The following five exercise configurations were performed in this order:
Hang power snatch with 20 kg load. The starting position was with the barbell rested on the distal (lower) third of the thighs and the hands gripping the bar outside the thighs. The end position was with arms extended above the head. Correct execution comprised one fluid movement of the bar and a stable ‘catch’ of the bar at the end position. A professional strength and conditioning coach oversaw all measurements, and repetitions deemed by him not to have met these criteria were excluded and repeated.Countermovement jump with ~50% of the load determined for moderate back squat (mean ± standard deviation: 34 ± 10 kg). Participants stood erect with the barbell on their shoulders (back squat position). In one fluid movement, participants descended (countermovement) by flexing the hips, knees, and ankles, then rebounded immediately into an explosive vertical jump. Depth of the countermovement was not standardized; however, when participants descended to a knee angle smaller than 90° (inspected visually by an investigator), repetitions were excluded and repeated with a shorter countermovement, so as to keep jumps as explosive as possible.Squat jump with ~50% of the load determined for moderate back squat (mean ± standard deviation: 34 ± 10 kg). With the barbell on shoulders (back squat position), participants descended to a knee angle of approximately 90° (inspected visually by an investigator). After this position had been held stable for ~2 s, an audible command was given to jump without any further countermovement. Repetitions with a slight countermovement (determined visually by the strength and conditioning coach) were excluded and repeated.Moderate back squat with the individual load determined during warm-up (65 ± 20 kg) to elicit ***v***_mean_ in the range of 0.7–0.8 m/s (mean ± standard deviation of actual values: 0.75 ± 0.05 m/s). With the barbell on the shoulders, participants descended in a controlled manner (although eccentric velocity was not standardized) by flexing the hips, knees, and ankles. Thereafter, they re-ascended by extension of the hips, knees, and angles with a maximal voluntary speed but no lift-off at the end of the extension. Depth of squats was not standardized, but participants were instructed to descend to a knee angle of 90° or further, but at most to the depth with thighs horizontal. Visual inspection by an investigator confirmed that squats were executed with a depth in this range. Sets with three or more repetitions lying outside the desired velocity range (based on ***v***_mean_ feedback from the GymAware device) were excluded and repeated with a corrected load.Heavy back squat with the individual load determined during warm-up (90 ± 20 kg) to elicit ***v***_mean_ of just under 0.5 m/s (actual values: 0.47 ± 0.05 m/s). These were performed in the same manner as the moderate back squats but with greater loads.


A uniform load was employed for the hang power snatch for simplicity’s sake and because the actual load used was not considered relevant to the research question. Rather, due to differences in participants’ abilities, the uniform load conveniently elicited a wider range of observed velocities for this exercise (compared to the other configurations, where individualized loads were used and the range of velocities was narrow), which we considered preferable for addressing the research question over a more ample range of feasible movement speeds. The two velocity realms for squats were chosen because they correspond to intensities of ~75–80% 1-RM, typically used for strength endurance training or training aimed at inducing hypertrophy, and ~90% 1-RM, typically used for improving maximal strength [19,20], respectively. Moreover, these velocity realms filled out the slower end of the spectrum of typical training velocities not covered by the other three exercises.

### 2.2. Data Collection

Four mobile VBT devices and one two-sided stationary VBT device, whose characteristics are summarized in Table 1, were connected to tablets (iPad Pro, Apple Inc., Cupertino, CA, USA, or Surface Go, Microsoft Corp., Redmond, WA, USA), which recorded and registered various parameters including ***v***_mean_ and ***v***_peak_ for each performed repetition. The devices were positioned on the barbell according to typical practice, as displayed in Figure 1.

Meanwhile, 3D motion capture data of the barbell’s trajectory were collected using eight infrared cameras (Vantage 5, Vicon Motion Systems Ltd., Oxford, UK) placed surrounding the participant and spherical reflective markers affixed at both ends of the barbell. A six-marker symmetrical arrangement at each bar end and a three-marker arrangement around the middle of the barbell facilitated the localization of the bar’s virtual left and right endpoints and midpoint, respectively. The Vicon cameras were controlled from an Antec WorkBoy desktop (Antec, Taipei, Taiwan) running Vicon Nexus software (version 2.9, Vicon Motion Systems Ltd., Oxford, UK). Sampling rate for the motion capture data was 100 Hz.

Following data collection, ***v***_mean_ and ***v***_peak_ values from all repetitions and VBT devices were extracted from the tablets and organized in a spreadsheet (Microsoft Excel, Microsoft Corp., Redmond, WA, USA) by participant, exercise, set, and repetition number.

Criterion parameters’ ***v***_mean_ and ***v***_peak_ were generated for each repetition by processing the raw 3D motion capture data (see Section 2.3), after which these too were organized by participant, exercise, set, and repetition number and then appended to the spreadsheet containing the other data.

### 2.3. Generation of Criterion Data

Criterion parameters’ ***v***_mean_ and ***v***_peak_ were generated for each repetition by processing the motion capture data according to the steps described in detail in the Appendix A (Appendix A). Briefly, the right and left ends of the barbell and their trajectories were determined using the six reflective markers placed on each of them. Based on these and the measured distance along the barbell where a given VBT device was attached, a virtual marker on the barbell representing the device’s measurement location was generated, along with its trajectory and velocity signal; velocity of this point was used as the criterion reference for that VBT device specifically. For each repetition and attachment point, concentric phases were identified automatically using an inclusive vertical velocity onset-threshold of 0 m/s for jumps and squats [9,18] or 0.05 m/s for hang power snatch. A slightly higher threshold was adopted for hang power snatch trials after graphical inspection of these revealed that a threshold of 0 m/s often designated a premature concentric phase onset (as the barbell crept upward before the actual pull began), thus leading to an underestimation of ***v***_mean_ by 0.3 m/s or more. In contrast, the more conservative threshold (0.05 m/s) might have overestimated ***v***_mean_ by only ~0.02 m/s or less (by missing the initial 0.01–0.02 s of the concentric phase). A threshold of 0 m/s was used to determine the end of the concentric phase for all exercises [9,18]. All repetitions were assessed graphically to ensure that concentric phases had been identified correctly. Finally, mean (***v***_mean_) and peak (***v***_peak_) resultant (scalar) velocity were calculated within this concentric phase.

### 2.4. Inspection and Exclusion of Data

For each VBT device and velocity parameter separately, a preliminary linear regression was generated, based on all available repetitions, relating device data and criterion data. From this, the standardized residual of each data point was computed (residual/standard deviation of all residuals) and data points with standardized residuals greater than 2 were inspected more closely [21]. Particularly, the integrity of the criterion data was scrutinized with the help of position–time and velocity–time plots of the device’s attachment point (generated from the motion-capture data). If the criterion data appeared questionable (for example due to a poorly definable concentric phase onset, or a particularly unusually shaped concentric velocity curve), the exclusion of the repetition was considered justifiable, and the data point was therefore excluded from analysis. This process was continued until each repetition either had a standardized residual of less than 2 or displayed no signs of errors in the criterion data. In the end, this process led to the justifiable exclusion of ~5% of all performed repetitions. All other repetitions were included in the statistical analyses.

### 2.5. Statistical Analyses

The validity of each VBT device was assessed for ***v***_mean_ and ***v***_peak_ separately using all included repetitions and the three-tier approach recommended by Hopkins [22] comprising (1) a Pearson’s correlation coefficient (*r*) (2) a calibration equation, and (3) the standard error of the estimate (SEE). The calibration equation assesses accuracy of measurement, whereas *r* and SEE assess precision. All statistics were calculated with customized Python scripts employing the SciPy library [23]. Because data from both criterion and practical measurement systems were subject to some random measurement error, calibration equations were generated using ordinary least product (OLP) regression [24]. Regression parameters (slope, intercept) and their 95% confidence limits were calculated based on methods described by Ludbrook [25]. The SEE was calculated manually from residuals of the OLP calibration equation as
(1)$$\mathrm{SEE}=\sqrt{\frac{1}{n-2}\sum_{i=1}^{n}{\left[Y_{i}-(a+bX_{i})\right]}^{2}}$$
where $X_{i}$ and $Y_{i}$ are the individual device and criterion data points, respectively, and *a* and *b* are the intercept and slope from the OLP regression [26]. Additionally, the SEE expressed as a percentage of the mean criterion value (SEE_pct_) was calculated. Correlation coefficients were interpreted with lower thresholds of 0.5, 0.7, and 0.9 for large, very large, and extremely large, respectively [21]. The absolute SEE was interpreted by supposing two meaningful thresholds: 0.1 m/s, which would be adequate for identifying a 30% velocity loss at relatively high loads [2], and 0.3 m/s, which would suffice for targeting specific goal-oriented training zones [19]. Thus, SEE less than 0.1 m/s and greater than 0.3 m/s was considered low and high, respectively, whereas SEE between 0.1 and 0.3 m/s was considered moderate. Proportional measurement bias was considered to exist if the 95% confidence limits of the calibration slope did not include 1, while a fixed measurement bias was considered to exist if the 95% confidence limits of the calibration intercept did not include 0 [27]. These statistics were run for each exercise separately and for all exercises pooled together. End statistics (*r*, SEE, SEE_pct_, slope, intercept) for Quantum were reported as the averaged end statistics from the two devices to ensure that these were based on a comparable number of data points as for the other devices. Based on each exercise separately, devices were ranked for Pearson’s *r*, SEE, and SEE_pct_, and these ranks were then summed to obtain an overall validity ranking of the VBT devices for the parameters ***v***_mean_ and ***v***_peak_.

For each of these statistical calculations, device data as they appeared in the mobile app were compared with the motion-capture data generated for the device’s own attachment point on the barbell. As such, the pure technological accuracy and precision of each device could be determined, independent of its location along the barbell. However, in addition, the same statistical analyses were performed comparing the outermost point of the barbell or a point just outside the typical grip width (0.43 m from bar end) with the barbell’s true midpoint. This yielded the isolated effect of attachment point itself, without regard to any VBT device.

## 3. Results

### 3.1. Data Set Description

In total, 724 repetitions were recorded from 14 participants performing five different free-weight exercise configurations (complete data set in Appendix A). Broken down by exercise configuration, 171, 139, 136, 140, and 138 repetitions were performed for hang power snatch, countermovement jump, squat jump, moderate back squat, and heavy back squat, respectively. Using the objective procedure for identifying outliers described in Section 2.4, 37 repetitions were excluded from ***v***_mean_ analyses, generally because the onset of the concentric phase could not be determined clearly in the criterion data. Further, 21 of the identified outlier repetitions were excluded from ***v***_peak_ analyses because of apparent errors in the criterion data. Moreover, each device occasionally failed to record a repetition for no apparent reason, in which case that repetition could not be analyzed for the given device. The resulting data set used for assessing validity is outlined in Table 2.

### 3.2. Validity of Devices

#### 3.2.1. Precision

The main indicators of precision for all devices and each exercise configuration, with characteristic velocity ranges, are displayed in Figure 2.

Overall precision rankings of the devices based on the sum of ranks from individual exercise configurations are displayed in Table 3 (***v***_mean_) and Table 4 (***v***_peak_).

#### 3.2.2. Accuracy

The regression equations for the individual exercises indicated that regression slopes differed depending on the exercise within any given device; this was the case for both ***v***_mean_ and ***v***_peak_. For ***v***_peak_, slopes varied between individual exercises rather uniformly, being generally steeper for low-velocity exercises and continually becoming flatter for higher-velocity exercises (Appendix A). Depending on device, the range of slopes was as small as 1.01–1.34 (Quantum) or as great as 0.97–1.63 (Vmaxpro). Generally, slopes for ***v***_mean_ varied to a lesser degree between exercises (Appendix A): as little as 0.98–1.15 (GymAware) or as much as 0.75–1.33 (Push). However, graphical inspection indicated that ***v***_mean_ regressions for hang power snatch were consistently left-shifted compared to those for other exercises (Appendix A), indicating systematic underestimation of ***v***_mean_ for this exercise by all VBT devices.

For this reason, the regression equations based on pooled exercise configurations were generated both with and without hang power snatch (Table 5, Appendix A). These revealed proportional bias for ***v***_mean_ with all devices, and for ***v***_peak_ with GymAware, Vmaxpro, and Push. Fixed bias was observed for ***v***_mean_ with Push and Flex and for ***v***_peak_ with all devices except Push.

### 3.3. Effect of Attachment Point

The correlation coefficients comparing the barbell midpoint to a point just outside the grip width were nearly perfect (>0.99), while SEE (0.04, 0.03 m/s for ***v***_mean_ and ***v***_peak_, respectively) and SEE_pct_ (3.1%, 1.7%) were small (Figure 3). Taking a virtual attachment point at the bar end, correlation coefficients remained essentially the same (≥0.98), and although SEE (0.06–0.05 m/s) and SEE_pct_ (4.6%, 2.7%) increased slightly, they remained quite small for both parameters.

## 4. Discussion

The current study assessed the pure technological validity of mobile and stationary VBT devices over a range of velocities and free-weight exercises, while also isolating the effect of device placement on velocity measurements. This study agrees with several others, that the GymAware is likely the most valid mobile VBT device on the market for ***v***_mean_ and ***v***_peak_, while also providing new data supporting its validity for ballistic and plyometric exercises. A novum in the current study is that the IMU device Vmaxpro competes very well with the GymAware in all tested velocity ranges and exercise types. Another new insight is that the stationary device Quantum sets the highest standard for determining ***v***_peak_, as well as a good validity for ***v***_mean_ for the wide variety of exercises and velocities tested. Finally, this study was the first to quantify and classify the likely influence of device attachment point on measurements of barbell velocity, revealing that discrepancies between mid-bar velocity and velocity at the bar end exist but are probably irrelevant in practical training settings.

Two of the tested devices, GymAware and Push, have been evaluated in a similar manner repeatedly [6,7,8,10,15,16,18,28,29], and both devices already have a body of literature supporting their validity for measuring barbell velocity for exercises such as back squat and squat jump (two of the exercises in the current study), as well as deadlift and bench press, with data quality generally better for the GymAware [4,7,8,29]. Comparing our results for GymAware with previous studies that used motion capture as their criterion measure, we observed similar, very high correlations (*r* of ~0.97) for ***v***_mean_ and ***v***_peak_ of back squats as was the case elsewhere [7,8,15]. Typical errors (SEE) we observed for ***v***_mean_ and ***v***_peak_ of back squats (~0.01 and ~0.03 m/s, respectively) and squat jumps (0.05 and 0.06 m/s, respectively) were very low and similar to those in the study of Mitter et al. [7]. For Push, correlations (*r* of ~0.8) and typical errors reported here for back squats are similar to the only other methodologically comparable study [7], again putting Push clearly behind GymAware in terms of validity. Whereas none of these studies included Olympic lifts or exercises with a stretch-shortening cycle, we investigated countermovement jump and hang power snatch. GymAware showed a high correlation and low SEE for both these exercises as well, whereas Push displayed only a moderate correlation for countermovement jump and an especially high SEE for both exercises. Moreover, GymAware failed least frequently (1% of the time) among all tested mobile devices to record a rep, whereas this was slightly more of an issue with Push (3%).

The other mobile devices tested in the current study have appeared seldom (Flex) [9,10] or not at all (Vmaxpro) in previous peer-reviewed literature. The mobile device Flex was similar to Push in terms of validity, displaying clearly poorer precision than the three best devices in the current study, especially for the more explosive movements. Although Flex measured ***v***_mean_ for non-ballistic back squats rather well (*r*: 0.84–0.94, SEE: 0.02–0.04), as was the case in the only other comparable study [9], its precision for the faster, ballistic and plyometric exercises, as well as for the parameter ***v***_peak_, which were assessed here for the first time, was rather poor compared to that of the three best devices in the study. Since precision indictors tended to be better for CMJ than for SJ, it appears that Flex struggles with high velocities, rather than with quick countermovements. This device also missed the greatest percentage of repetitions (10%) among all tested devices. On the other hand, Vmaxpro displayed consistently high precision across the various exercise forms and velocity ranges, generally remaining close behind the best-ranked devices in the study. Further, having missed 5% of all repetitions, it was only slightly poorer than average in this regard. This being the first peer-reviewed study to classify Vmaxpro among the more valid mobile VBT devices on the market, follow-up studies are warranted to solidify our findings.

As a stationary device, the 1080 Quantum is in a different category than the other tested (mobile) devices. It seemed nonetheless important to assess the Quantum’s validity due to its increasing popularity and use for VBT. Generally, it measured up very well for ***v***_mean_ for all exercises except the hang power snatch, while displaying better precision for ***v***_peak_ than all tested mobile devices. Judging by our efforts in generating the criterion velocity data (from motion capture), identifying the onset and end of the concentric phase was most challenging for the hang power snatch due to differences in experience and explosive strength among participants. Slight irregularities in movement around the ends of the concentric phase may have been further exacerbated or amplified for the Quantum, since its ropes where nearly completely reeled out in the current study’s setting, and thus slacker (e.g., compared to GymAware’s cable). This may have detracted from Quantum’s precision for ***v***_mean_ during hang power snatch, whereas precision indicators were nonetheless quite good for other exercises. Further, for the parameter ***v***_peak_, which occurs later in the concentric phase where ropes or cables are most certainly taut, Quantum set the highest benchmark among all tested devices and parameters, displaying nearly perfect agreement with the criterion measures at both ends of the velocity spectrum. Furthermore, it failed to record a repetition less frequently (<1% of the time) than all mobile devices.

Regarding device accuracy, the individual-exercise calibration equations for ***v***_mean_ indicate that velocities can be compared between various squats and vertical jumps within the same device, whereas ***v***_mean_ from the hang power snatch may be systematically underestimated due to fixed measurement biases of as little as ~0.1 m/s (GymAware, Vmaxpro) or of >0.3 m/s (Push, Quantum). While processing the criterion data, it became apparent that the concentric phase onset for the hang power snatch is often difficult to identify precisely. Because participants often adjusted the bar position upward to find their preferred starting position before initiating the actual explosive movement, indiscriminately using the lowest position or the first time point of upward movement (velocity >0 m/s) as the onset of the concentric phase clearly underestimated ***v***_mean_ in such cases. We therefore used a threshold of 0.05 m/s, considering systematic slight overestimation of ***v***_mean_ (by <0.02 m/s) preferable to a random underestimation thereof. In any case, for exercises such as the hang power snatch, where the starting position may be difficult to identify, and perhaps particularly with weightlifting beginners whose technique varies more, the mobile devices face the same challenge. As it seems from the current data, device algorithms tend to err on the side of underestimation of ***v***_mean_ in this situation.

Regardless of systematic and slight over- or underestimations, practitioners might agree that for the purpose of monitoring training, as long as one decides upon a device to use and sticks with it, precision is more important than accuracy. Further, comparing velocity within the same exercise is more important than comparing between exercises; therefore, SEE for individual exercises is perhaps most important for assessing the validity of a devices, whereas the slopes and intercepts of calibration equations, as well as discrepancies in these between exercises, could be considered less relevant. Thus, the rankings in Table 3 and Table 4 suffice as a gauge of overall validity of the tested devices.

A merit of the current study is that our criterion velocity data were taken from the exact location on the barbell where the tested device was attached, as has been the case for some previous studies as well [7,9,18]. This is in contrast to other studies, which used mid-bar velocity (when motion capture was used as the criterion) or some other random point on the bar (such as where another VBT devices was used as the criterion), while either assuming or ensuring parallel bar movement [8,10,15,16,17]. We determined the velocity of the true attachment point for each device in order assess devices’ technological validity more fairly. Nonetheless, it has been shown [14] or inferred [29,30] that, during free-weight exercises, the barbell does not always move perfectly parallel.

For this reason, we also quantified the isolated effect device placement can have on velocity parameters, which was a unique feature of the current study. When the bar departs from parallel movement, the velocity of the leading end is exaggerated while that of the trailing end is diminished. This phenomenon would be expected to be stronger the farther away a device is attached from the bar’s midpoint [14], and to affect ***v***_peak_ (because it is an instantaneous value) more than ***v***_mean_, (since tipping occurring at some instant during the concentric phase is likely compensated for by the end of the movement). However, if tipping confounds the detection of the onset or conclusion of the concentric phase, it could affect ***v***_mean_ as well. Appleby et al. have previously provided evidence that barbell displacement differs depending on the point of reference and due to barbell bending, with increasing differences the farther apart along the barbell measurements are made [2]. The current study extends on the work of Appleby et al. by quantifying the measurement error for velocity parameters based purely on the site of measurement along the barbell. Our results reveal maximal typical measurement errors (SEE) of only 0.06 m/s (for the parameter ***v***_peak_, when a device is attached to the bar end) compared the middle of the bar. Further, discrepancies between mid-bar and bar-end velocities parameters typically vary by only 3–5% (i.e., by only ~0.1 m/s at 2 m/s). In all likelihood, such errors are too small to confound velocity-based training quality or progress. However, where VBT devices are used for field tests or research purposes, the location of device attachment might need to be considered more carefully.

## 5. Conclusions

This study reconfirmed the high validity of the GymAware, expanding findings to previously unexplored ballistic and plyometric exercises. The stationary 1080 Quantum system also displayed excellent validity characteristics for the wide range of free-weight exercises and velocities, and these two missed a repetition least frequently among the devices tested. While these two devices appear to set the standard for measurement precision, Quantum is not mobile and GymAware is substantially more expensive than its mobile competitors. With this in mind, it was somewhat surprising that the second-best mobile device in the current study, only slightly behind GymAware, was also the cheapest: Vmaxpro. The Vmaxpro is also an IMU and the smallest of the tested devices, making it quicker to install and more versatile than the others. This finding, should it be confirmed in follow-up studies, is important not only for consumers on a small budget but also for those interested in a minimalistic, simple-to-use, and valid VBT device. Finally, this study showed that effects of device attachment location are small and, while perhaps relevant in research setting, likely negligible during training practice.

Taking a step back, validity alone may not be the only important criterion when choosing a VBT device. A device’s tendency to miss repetitions, for which descriptive data are provided by the current study, could be a differentiating factor. Moreover, the usability, range of functions, and supported operating systems of the app used to control the device and display and manage data are also important factors for consumers to consider in order to make an informed decision about the most suitable device for them.

## Figures and Tables

**Figure 1 sports-09-00123-f001:**
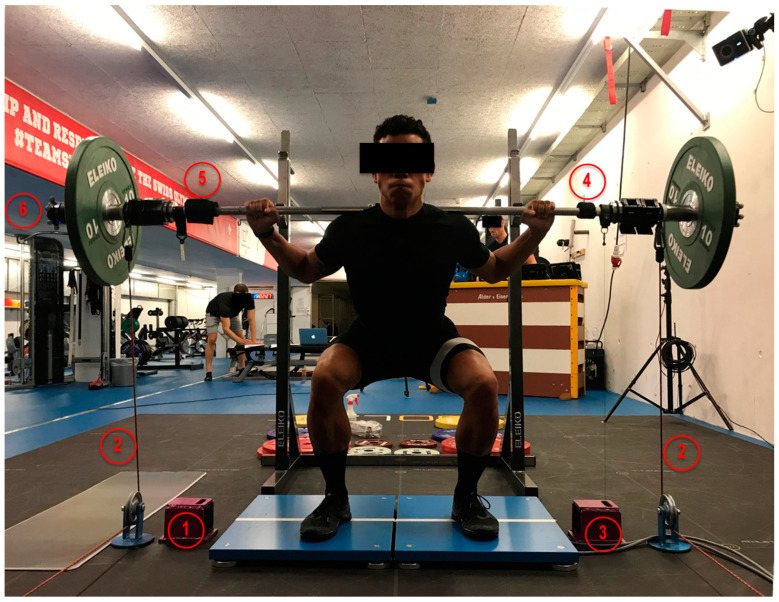
Study setup and locations of the tested devices on the barbell. (1) GymAware, (2) 1080 Quantum, (3) outdated linear position transducer not included in this study, (4) Vmaxpro, (5) Push, (6) Flex. A dual force plate was also in place for collecting data not addressed in the current study.

**Figure 2 sports-09-00123-f002:**
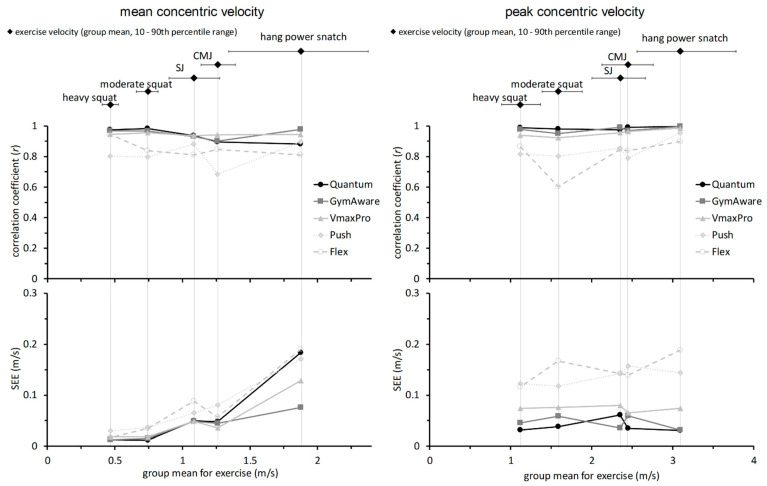
Validity indicators by device and exercise configuration for ***v***_mean_ (**left** panel) and ***v***_peak_ (**right** panel). The black diamonds at the top and their horizontal bars indicate the group average and 10th–90th percentile range, respectively, for the corresponding velocity parameter and the indicated exercise configuration. SEE: standard error of the estimate from least-products linear regression.

**Figure 3 sports-09-00123-f003:**
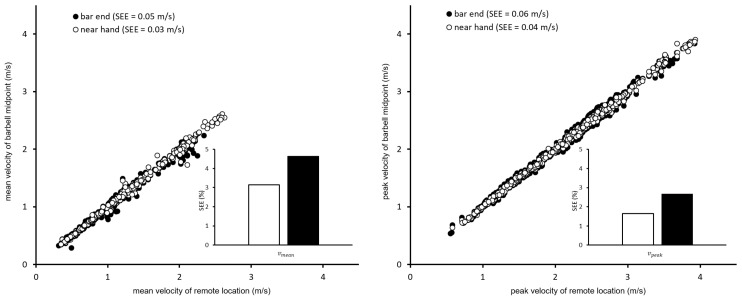
Scatter plots of mean (**left** panel) and peak (**right** panel) velocity for remote device locations versus true barbell midpoint. Bar plots convey percentage errors in velocity measurements compared to the barbell midpoint due to measurement location along the barbell. SEE: standard error of the estimate from least-products linear regression. ***v***_mean_, ***v***_peak_: mean and peak concentric barbell velocity, respectively.

**Table 1 sports-09-00123-t001:** Characteristics of tested devices.

Device	Version	Type	Sampling Rate (Hz)	Attachment Point
GymAware PowerTool ^a^	f: 16.0.1.a024, a: 2.8.8	LPT	50	363 mm from right bar end
Vmaxpro VMP5 ^b^	f: 5.2.0	IMU	200	425 mm from left bar end
Push ^c^	a: 7.4.0	IMU	200	415 from right bar end
Flex ^a^	f: 2710, a: 1.9.32	Laser	50	right bar end
1080 Quantum ^d^	a: 5.0.4.2	LPT	333	205 mm from left and right bar ends

LPT: linear position transducer. IMU: inertial measurement unit. f: firmware version. a: mobile application version. ^a^ PowerTool, Kinetic Performance Technology, Mitchell, Australia. ^b^ VMP5, Blaumann & Meyer Sports Technology, Magdeburg, Germany. ^c^ Band 2.0, PUSH Inc., Toronto, Canada. ^d^ 1080 Motion, Lidingö, Sweden.

**Table 2 sports-09-00123-t002:** Description of the data set.

Device	Missed (*n*)	Missed (%)	Analyzed (*n*), ***v***_mean_	Analyzed (*n*), ***v***_peak_
GymAware	4 (0–3)	1% (0–2%)	687 (128–147)	701 (129–157)
Vmaxpro	35 (0–30)	5% (0–22%)	652 (103–146)	668 (104–158)
Quantum	3 (0–2)	<1% (<1%)	686 (125–150)	705 (130–160)
Push	25 (0–18)	3% (0–11%)	665 (128–138)	682 (132–142)
Flex	73 (0–58)	10% (0- 34%)	622 (96–140)	630 (101–140)

Data are displayed as the total numbers (*n*) or percentages (%) of repetitions over all five exercise configurations, with the range of repetitions per exercise configuration in parentheses. A total of 724 (138–171 per exercise) repetitions were performed during data collection, of which 37 (0–23 per exercise) were justifiably excluded from analyses of ***v***_mean_ and 21 (0–12 per exercise) were justifiably excluded from analyses of ***v***_peak_. See text for details on justifiable exclusions. Missed repetitions were those not recorded by the VBT device for no apparent reason.

**Table 3 sports-09-00123-t003:** Precision indicators and device rankings for the parameter ***v***_mean_.

Device	Rank	Pearson’s *r*	SEE (m/s)	SEE_pct_ (%)
GymAware	1	0.99 (0.90–0.98)	0.06 (0.01–0.08)	5.4 (2.0–4.5)
Vmaxpro	2	0.99 (0.94–0.96)	0.08 (0.02–0.13)	7.0 (2.4–6.8)
Quantum	3	0.97 (0.88–0.98)	0.13 (0.01–0.18)	11.7 (1.6–9.8)
Flex	4	0.96 (0.81–0.94)	0.12 (0.02–0.19)	11.2 (3.6–10.1)
Push	5	0.97 (0.69–0.90)	0.12 (0.03–0.17)	11.0 (5.0–9.1)

Overall rankings are based on summed ranks of devices for the three displayed statistics and the five individual exercise configurations. Statistics outside parentheses come from all exercises pooled together (*n*: 622–687 repetitions), while the range of values obtained from the individual exercises (*n*: 96–150 repetitions, see Table 2) are shown in parentheses. SEE: standard error of the estimate from least-products linear regression.

**Table 4 sports-09-00123-t004:** Precision indicators and device rankings for the parameter ***v***_peak_.

Device	Rank	Pearson’s *r*	SEE (m/s)	SEE_pct_ (%)
Quantum	1	1.00 (0.97–1.00)	0.07 (0.03–0.06)	3.2 (1.0–2.8)
GymAware	2	0.99 (0.95–1.00)	0.08 (0.03–0.06)	3.6 (1.0–4.0)
Vmaxpro	3	0.99 (0.92–0.99)	0.11 (0.07–0.08)	5.2 (2.4–6.6)
Flex	4	0.96 (0.60–0.90)	0.18 (0.12–0.19)	8.6 (5.7–10.5)
Push	5	0.98 (0.79–0.96)	0.15 (0.12–0.16)	7.1 (4.7–11.0)

Overall rankings are based on summed ranks of devices for the three displayed statistics and the five individual exercise configurations. Statistics outside parentheses come from all exercises pooled together (*n*: 630–705 repetitions), while the range of values obtained from the individual exercises (*n*: 101–160 repetitions, see Table 2) are shown in parentheses. SEE: standard error of the estimate from least-products linear regression.

**Table 5 sports-09-00123-t005:** Complete calibration equation parameters with confidence limits (c.l.).

Device	Parameter	Slope (95% c.l.)	Intercept (95% c.l.)
GymAware	** *v* ** _mean_ ^a^	0.98 (0.97, 0.99)	0.00 (−0.01, 0.00)
	** *v* ** _mean_ ^b^	1.08 (1.07, 1.09)	−0.10 (−0.11, −0.08)
	** *v* ** _peak_	0.97 (0.96, 0.98)	0.22 (0.20, 0.24)
Vmaxpro	** *v* ** _mean_ ^a^	1.05 (1.04, 1.06)	−0.01 (−0.02, 0.00)
	** *v* ** _mean_ ^b^	1.14 (1.12, 1.16)	−0.09 (−0.11, −0.07)
	** *v* ** _peak_	0.94 (0.92, 0.95)	0.31 (0.28, 0.34)
Quantum	** *v* ** _mean_ ^a^	0.97 (0.96, 0.98)	0.01 (0.00, 0.02)
	** *v* ** _mean_ ^b^	1.16 (1.13, 1.19)	−0.16 (−0.19, −0.12)
	** *v* ** _peak_	0.99 (0.98, 1.00)	0.19 (0.17, 0.21)
Push	** *v* ** _mean_ ^a^	1.09 (1.07, 1.11)	−0.03 (−0.05, −0.02)
	** *v* ** _mean_ ^b^	1.22 (1.19, 1.25)	−0.14 (−0.17, −0.10)
	** *v* ** _peak_	1.17 (1.15, 1.20)	0.03 (−0.02, 0.08)
Flex	** *v* ** _mean_ ^a^	0.98 (0.96, 0.99)	0.01 (0.00, 0.02)
	** *v* ** _mean_ ^b^	1.23 (1.19, 1.27)	−0.22 (−0.27, −0.18)
	** *v* ** _peak_	0.98 (0.95, 1.01)	0.28 (0.22, 0.33)

Slopes were generated using least-products linear regression. Equation parameters for ***v***_mean_^a^ were generated without hang power snatch, for which there was possibly a systematic underestimation of mean velocity by mobile devices (Appendix A, discussion in text). Equation parameters for ***v***_mean_^b^ and ***v***_peak_ are based on all exercises (including hang power snatch).

## Data Availability

The resulting data set is available under the link for Appendix A.

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
