# Peer review of "Validity and Effects of Placement of Velocity-Based Training Devices"

_sports, 2021, doi:10.3390/sports9090123_

Round 1

Reviewer 1 Report

General Comments:

The authors need to provide a stronger rationale within the introduction as to why this research project is necessary.  Currently, a small paragraph is included, but it does not build off of previous paragraphs.  The methods section requires a lot of work.  Currently, the study is not replicable in its current form.  Please see the specific comments below.

Specific Comments:

Line 50: Change "the fewest of studies" to "a handful of studies".

Line 64: References should be provided to support the first portion of this statement (i.e. following 'valid').

Lines 71-76: This section and part of the introduction needs to be revised to further build the research question.  At present, this paragraph does not come with much support and is not tied into other paragraphs.  The authors need to provide a rationale as to why the placement of devices is important.

Line 83: The authors should include a hypothesis.

Line 90: Based on the training status of the participants, can the authors be confident that exercise technique did not play a role?  Individuals can modify their technique and achieve a faster/slower velocity.

 Line 96: Were these loads chosen by the authors?  If so, the loads should be included for replication purposes.  How many repetitions/trials did it take to reach these values?  

Lines 98-100: What was the rationale behind using these velocities?  Further support is needed here.

Lines 102-118: Were the participants familiarized with performing the movements with all of devices on the bar prior to testing repetitions?  This many devices can be a distraction.

Line 106: This is confusing.  It is unclear how many repetitions were performed, how many sets were performed, how the loads for the power snatch (hang power snatch/power snatch from the knee), countermovement jump, and squat jump were determined, the technique that was used for each exercise, how much rest was provided between trials (assuming multiple trials were performed), etc.  At present, this study is not replicable.

Line 108: This exercises as noted above should be classified as a power snatch from the knee based on the starting position.

Lines 116-118: Here the authors mention 'repetitions', but it is unclear how many were performed within each set.  Perhaps there is some confusion on terminology.

A brief description of the technique used for each exercise must be included.  For example, readers should know the depth of the squats (assuming it is a back squat based on the Figure), the instructions used for each exercises, and the procedures if another repetition needed to be performed based on technique (e.g. power snatch with beginners).

Line 121: The company city, state, and country should be included as is common in the literature.

Line 152: Why were different thresholds used?  This requires support from previous literature or a sound rationale.

Line 171: What data were used for each participant?  Was the average data used for each participant or was it the best trial?  This is currently unclear.  CV is a measure of absolute reliability.  The authors should consider including a measure of relative reliability using intraclass correlation coefficients in addition to the CV.  Furthermore, the authors should provide interpretation scales to show how correlation, reliability, etc. were interpreted.

Based on the comments above, no additional feedback was provided as the suggested changes may alter the reporting of the results and discussion.

Author Response

General Comments:

The authors need to provide a stronger rationale within the introduction as to why this research project is necessary.  Currently, a small paragraph is included, but it does not build off of previous paragraphs.  The methods section requires a lot of work.  Currently, the study is not replicable in its current form.  Please see the specific comments below.

Dear reviewer

Thank you for taking the time to read the manuscript and formulate your critiques and comments. Several of your points have helped improve the paper.

The authors

Specific Comments:

Line 50: Change "the fewest of studies" to "a handful of studies".

  • I have made the change according to your recommendation.

Line 64: References should be provided to support the first portion of this statement (i.e. following 'valid').

  • This statement has been improved and references are now included.

Lines 71-76: This section and part of the introduction needs to be revised to further build the research question.  At present, this paragraph does not come with much support and is not tied into other paragraphs.  The authors need to provide a rationale as to why the placement of devices is important.

  • The introduction has been revised to flow more logically and provide better rationale for assessing device placement.

Line 83: The authors should include a hypothesis.

  • Good idea. The following hypothesis has been added (lines 106 – 108): ‘We hypothesized that each devices would provide valid velocity measurements for its own attachment point, but that attachment locations closer the barbell end would amplify discrepancies between device measures and barbell mid-point velocity.’

Line 90: Based on the training status of the participants, can the authors be confident that exercise technique did not play a role?  Individuals can modify their technique and achieve a faster/slower velocity.

  • Much to the contrary, we assume that (intra- and intersubject variations in) technique indeed played a role, a point which now receives more attention in the Methods section of the revised manuscript (see section 2.1). To be sure, modifications in technique affect velocity as you write, and this is precisely the challenge that device developers face, as they cannot control the technique of consumers or expect them to train without some technical errors. To appease your concern, the following additions have been made to the text:
    • The wide range of ability and technical savvy within the subject cohort was intentional. In contrast to a cohort with abundant free weigh training experience, this variety was sup-posed to better reflect the ability levels represented by consumers of the tested devices. (lines 117 – 120)
    • [during the warm-up] each subject did a few repetitions of each tested exercise under the observation of (and if necessary abiding to the corrections of) one of the investigators to ensure they understood how each exercise was to be performed technically. (lines 124 – 126)
    • … A professional strength and conditioning coach oversaw all measurements… (lines 149 – 150)

 Line 96: Were these loads chosen by the authors?  If so, the loads should be included for replication purposes.  How many repetitions/trials did it take to reach these values?  

  • Loads for the squat measurments during warm-up were suggested by an investigator (as now stated in the text), in order to efficiently deduce the loads needed for the desired velocity realms for squats actual measurments. Although the actual loads used as well as the number of sets/reps required to obtain these loads, are irrelevant to the research question (which only addresses velocity measurements), the mean/s.d. loads are now included in the bulleted points in this section.

Lines 98-100: What was the rationale behind using these velocities?  Further support is needed here.

  • After consulting another investigator on this point, I have now re-specified the targeted velocity realm for the ‘moderate squat’ configuration (line 131), this being 0.5 – 0.85 m/s (rather than ‘~0.8 m/s’ as previously stated). Regarding the rationale for these realms, an explanation has been added (lines 189 – 194).

Lines 102-118: Were the participants familiarized with performing the movements with all of devices on the bar prior to testing repetitions?  This many devices can be a distraction.

  • No, subjects were not familiarized with performing the movement with all the devices on the bar. However, as the aim of the study was solely to assess the technology not the humans performing the movements, we conclude that, if indeed subjects were distracted by the multiple devices, this would have irrelevant to the study design and study results.

Line 106: This is confusing.  It is unclear how many repetitions were performed, how many sets were performed, how the loads for the power snatch (hang power snatch/power snatch from the knee), countermovement jump, and squat jump were determined, the technique that was used for each exercise, how much rest was provided between trials (assuming multiple trials were performed), etc.  At present, this study is not replicable.

  • ‘starting with the fastest and ending with the slowest’ has been removed from the text. This was meant to the first exercise was that with the highest speed and the last exercise that with the lowest speed, but it seems to have led to confusion. That the exercise configurations were performed in this order should remain clear from line 144.
  • To your comment, the number of reps per set (five) is now specified in the text. The number of sets per exercise configuration (1 – 2) is stated in line 138. The loads for the hang power snatch, were the same for all participants (20 kg, see bulleted point). This was done for simplicity’s sake and because the actual load used was not considered relevant for the research question. Moreover, as is visible from the range bars in Figure 3, using a uniform load for this exercise led to a wider range of observed velocities, which we consider preferable for addressing the research question, than was the case for the other exercise configurations, where individualized loads were used. This explanation has been added along with the rationale for the squat velocity realms (see lines 183 - 194). To reemphasize the point, we were investigating velocity measurments at various speed. The loads were merely a means to elicit these velocities. Thus, the actual loads used were widely irrelevant to the research question (even in regards to replicating the study, unless the study were to be replicated with exactly the same participants having undergone no changes in their physical condition). Nonetheless, loads are provided in the revised manuscript (they may be considered relevant when considering the effects of bar placement because of bar bending).

Line 108: This exercises as noted above should be classified as a power snatch from the knee based on the starting position.

  • You’re right, power snatch was not the correct name since the barbell didn’t start on the floor. We figure even more fitting than ‘power snatch from the knee’ is ‘hang power snatch.’ The term has been substituted in the text and Figure 3. Also, more detailed descriptions of all exercises have been added.

Lines 116-118: Here the authors mention 'repetitions', but it is unclear how many were performed within each set.  Perhaps there is some confusion on terminology.

  • In response to this and a comment from another reviewer, this sentence has been extended and improved: ‘To best reflect real-world usage of the VBT devices, eccentric velocity ( - 0.37 ± 0.14 m/s) was not standardized or manipulated. However, participants did reset to a stable, standing po-sition between reps, rather than performing the reps within a set continuously.’ (lines 140 – 143). The number of reps per set should now be clear from line 138.

A brief description of the technique used for each exercise must be included.  For example, readers should know the depth of the squats (assuming it is a back squat based on the Figure), the instructions used for each exercises, and the procedures if another repetition needed to be performed based on technique (e.g. power snatch with beginners).

  • Detailed descriptions have been added for the five exercises configurations. To your point regarding squat depth, a range but not a particular depth was given, as is now stated in the text. Regarding valid and invalid reps, specifically for hang power snatch, details are now provided.

Line 121: The company city, state, and country should be included as is common in the literature.

  • These details have been added to the text.

Line 152: Why were different thresholds used?  This requires support from previous literature or a sound rationale.

  • The rationale for using 0.05 m/s rather than 0 for the hang power snatch has been added to the text. There are no previous references for this because no previous studies have used motion capture and this exercise in combination. Thompson et al. (2020) used motion capture with the power clean exercise, but there the bar begins on the floor and moves clearly upward when the pull begins. With the hang power snatch, people sometimes tend to creep upward with the barbell (to their preferred starting position) before initiating the explosive pull. Although a threshold higher than 0 may have led to a very slight systematic overestimation of vmean (~0.02 m/s in simulations), we deemed this preferable to a random, not seldom underestimation of vmean (by >0.3 m/s) observed in our data when a threshold of 0 m/s was tried.

Line 171: What data were used for each participant?  Was the average data used for each participant or was it the best trial?  This is currently unclear.  CV is a measure of absolute reliability.  The authors should consider including a measure of relative reliability using intraclass correlation coefficients in addition to the CV.  Furthermore, the authors should provide interpretation scales to show how correlation, reliability, etc. were interpreted.

  • All repetitions were included in the analyses (i.e., without averaging or selecting repetitions by set). New additions to the text in section 2.5, along with the data in Table 2, should make this clear in the revised manuscript.
  • Thanks for the point about CV as it led me to take a closer look at this statistic in Hopkins’s spreadsheet. It turns out the CV I report is not what I thought it was (degree of variation of differences between the device and the criterion measure), rather it is essentially the SEE as a percent (of the predicted, i.e. criterion value). The parameter has been rebranded in the revised manuscript as SEEpct. Further, because this is somewhat redundant to with the absolute SEE, I’ve removed some referrals to it in the text, removed it from Figures 3 and 4, while leaving it in Tables 3 and 4.
  • Your suggestion to include the ICC is good, however, as I looked into this, there were essentially no differences between the ICC and Pearson’s r. Therefore, I see no value to including it.
  • Thresholds for the interpretation of correlation coefficients have been adopted from Hopkins and are now included in the manuscript (lines 202 – 203). Lacking any previous reference for categorizing absolute SEE, we’ve proposed a smallest meaningful difference of 0.1 m/s and based interpretations off of this (lines 267 – 272).

Based on the comments above, no additional feedback was provided as the suggested changes may alter the reporting of the results and discussion.

Reviewer 2 Report

This study aimed to assess and compare the validity of several mobile and one stationary velocity-based training device for measuring mean and peak concentric barbell velocity over a range of velocities and exercises. I would like to congratulate the authors for the magnificent work done on a current topic such as velocity-based strength training. Research on the validity of the different commercially available devices for measuring barbell velocity is still needed in the scientific literature. However, some small considerations should be taken into account before recommending acceptance of this work.

Abstract. To obtain a more detailed view, it would be convenient to reduce the information provided for the background (lines 8-11) and give more detail about the results and conclusions of the present study.

Lines 97, 264, and 276. Change "GymAwere" by "GymAware". Please check it throughout the text.

Line 107. Were 5 or 4 different exercises evaluated? I understand that the squat exercise is the same regardless of the load used.

Lines 108-114.  Provide a more detailed description of the characteristics of each exercise. Was the velocity of the descending phase of each exercise controlled? Please clarify.

Lines 116-117. Was the number of reps self-selected in each set?

Figure 1. Was a dual force platform also used? Please clarify in the method because the data were not included in the present study.

Line 211. Were the data of 11 or 14 subjects (i.e., 11 men and 3 women) reported?

Figure 3. Sorry, but without the legend, I don't know what the different symbols (circles, squares, diamonds, etc.) correspond to.

Lines 276. Compare the results observed in the present study with those previously reported in the literature (at least for the same exercises) and do not only refer to recent reviews.

Lines 332-335. I miss some references here. In addition, the authors should mention the work Appleby et al. (2020).

https://pubmed.ncbi.nlm.nih.gov/33105362/

Author Response

This study aimed to assess and compare the validity of several mobile and one stationary velocity-based training device for measuring mean and peak concentric barbell velocity over a range of velocities and exercises. I would like to congratulate the authors for the magnificent work done on a current topic such as velocity-based strength training. Research on the validity of the different commercially available devices for measuring barbell velocity is still needed in the scientific literature. However, some small considerations should be taken into account before recommending acceptance of this work.

Abstract. To obtain a more detailed view, it would be convenient to reduce the information provided for the background (lines 8-11) and give more detail about the results and conclusions of the present study.

  • More details have been added to the abstract.

Lines 97, 264, and 276. Change "GymAwere" by "GymAware". Please check it throughout the text.

  • Correction made to auto spell-check mistake

Line 107. Were 5 or 4 different exercises evaluated? I understand that the squat exercise is the same regardless of the load used.

  • You’re right. There were indeed four different exercise, but 2 different loading conditions for squats. In order to avoid confusion without being overly complex, I’ve changed ‘five exercises’ to ‘five exercise configurations.’

Lines 108-114.  Provide a more detailed description of the characteristics of each exercise. Was the velocity of the descending phase of each exercise controlled? Please clarify.

  • Detailed descriptions have been added for the five exercises configurations. Included in these is the declaration that the descending phase was performed ‘in a controlled manner’ (i.e., not explosive); however eccentric velocity was not a ‘controlled’ variable (in the sense of standardized), in order to reflect real-world conditions

Lines 116-117. Was the number of reps self-selected in each set?

  • I didn’t realize this detail was missing. Sets were of 5 reps each, as is now stated in line 138

Figure 1. Was a dual force platform also used? Please clarify in the method because the data were not included in the present study.

  • Yes, that is a force plate in the picture. To avoid confusion, I’ve amended the figure caption with ‘A dual force plate was also in place for collecting data not addressed in the current study.’

Line 211. Were the data of 11 or 14 subjects (i.e., 11 men and 3 women) reported?

  • They were from all 14 subjects (men and women combined). I’ve now corrected this error.

Figure 3. Sorry, but without the legend, I don't know what the different symbols (circles, squares, diamonds, etc.) correspond to.

  • Of course not; thanks for noticing. I’ve now corrected the figure and added the legend.

Lines 276. Compare the results observed in the present study with those previously reported in the literature (at least for the same exercises) and do not only refer to recent reviews.

  • Several new references were added to the manuscript as well as direct comparisons with the (comparable) results from those studies.

Lines 332-335. I miss some references here. In addition, the authors should mention the work Appleby et al. (2020).

  • Thanks for this comment. We had missed the paper of Appleby et al., but have now included it, which also helps build the rationale for the aspect of our study regarding device placement.

Reviewer 3 Report

Thank you for giving me the opportunity to review your manuscript. You have undertaken an extensive study to explore possible limitations with the way barbell velocity is recorded and I think that this work has a lot of really good potential and will be of interest to the readers of this journal. However, I have grave concerns about the statistical analysis of your data and recommend that you reanalyse them using least products regression or suitable alternatives (Deming regression for normally distributed data and Passing-Bablok for non-normally distributed data). These tests enable you to identify whether there is any fixed or proportional bias in your 'alternative' methods and so how well 'actual' velocity could be predicted by these methods. Additionally, it must be remembered that one of the assumptions of least squares regression is that the criterion method's data will not be noisy. Of course, this isn't the case. Least products regression accounts for this.

Additionally, I have some suggestions for minor revisions that I provide below:

Line 30 and throughout: please replace 'allows' with 'enables'; 'allows' suggests that permission is needed and this isn't the case.

Line 45: please replace 'suppose' with 'suggest'.

Line 47: please replace 'explosion' with something like 'increased popularity'.

Line 48: please replace 'pop up on the market' with something like 'continue to become commercially available'.

Line 49: please replace 'lopsided' with something like 'biased'.

Method section: please refer to either 'subjects' or 'participants' consistently throughout.

Line 117: please replace 'technical savvy' with something like 'technical competency'.

Line 122: please replace 'the vicinity' with something like 'attending one test session'.

Line 126: please replace 'he or she' with 'they'.

Line 137: with regards to how much load this device adds to the barbell, is it 2.5 kg per side? Regardless, please clarify this here.

Throughout: please replace 'rep' and 'reps' with 'repetition' and 'repetitions' throughout. 

Author Response

Thank you for giving me the opportunity to review your manuscript. You have undertaken an extensive study to explore possible limitations with the way barbell velocity is recorded and I think that this work has a lot of really good potential and will be of interest to the readers of this journal. However, I have grave concerns about the statistical analysis of your data and recommend that you reanalyse them using least products regression or suitable alternatives (Deming regression for normally distributed data and Passing-Bablok for non-normally distributed data). These tests enable you to identify whether there is any fixed or proportional bias in your 'alternative' methods and so how well 'actual' velocity could be predicted by these methods. Additionally, it must be remembered that one of the assumptions of least squares regression is that the criterion method's data will not be noisy. Of course, this isn't the case. Least products regression accounts for this.

Thank you for your review and especially this very helpful suggestion regarding the regression method. Indeed, we hadn’t thought to account for the random error in the criterion measurement and have re-run the statistics using the least-products regression method (as described in the cited reference from Ludbrook). In general this led to statistical indicators of validity being better than in the previous manuscript version. We even experimented with doing a weighted regression (with residuals of the unweighted regression serving as sigma), but there were virtually no differences from the unweighted regression; therefore we opted in the end for the more straightforward unweighted least-products regression.

Additionally, I have some suggestions for minor revisions that I provide below:

Line 30 and throughout: please replace 'allows' with 'enables'; 'allows' suggests that permission is needed and this isn't the case.

The suggested change has been made.

Line 45: please replace 'suppose' with 'suggest'.

The suggested change has been made.

Line 47: please replace 'explosion' with something like 'increased popularity'.

‘explosion’ has been replaced with ‘abundance’

Line 48: please replace 'pop up on the market' with something like 'continue to become commercially available'.

The suggested change has been made.

Line 49: please replace 'lopsided' with something like 'biased'.

The suggested change has been made.

Method section: please refer to either 'subjects' or 'participants' consistently throughout.

The suggested change has been made.

Line 117: please replace 'technical savvy' with something like 'technical competency'.

The suggested change has been made.

Line 122: please replace 'the vicinity' with something like 'attending one test session'.

The suggested change has been made.

Line 126: please replace 'he or she' with 'they'.

The suggested change has been made.

Line 137: with regards to how much load this device adds to the barbell, is it 2.5 kg per side? Regardless, please clarify this here.

The text has been revised as follows to be more clear about how the minimal resistance of 20 kg came to be:

two motor-controlled cable-pull devices (1080 Quantum, 1080 Motion, Lidingö, Sweden) were attached to either end of the barbell, each providing a resistance of 2.5 kg (5 kg total). As such, the minimal additional load was 20 kg.

Throughout: please replace 'rep' and 'reps' with 'repetition' and 'repetitions' throughout. 

The suggested change has been made.

Round 2

Reviewer 1 Report

While the authors have addressed most of the original minor concerns, there are still methodological flaws that, in this reviewer's opinion, prevent this study from being published (e.g. rationale for loading, velocities, exercise technique, statistics, etc.) 

Author Response

With the second revision of the manuscript, major changes to the statistical procedures have been made, namely a least-products regression method is now employed (rather than the non-parametric Theil-Sen method used in the previous version) in order to better account for random errors in the criterion measurments. This led generally to better validity of all tested devices. Rationale for velocities (widest realistic range of velocities encountered in real-world situations), loading (loads which produce those velocities at maximal intensity for the actual study participants), and exercise technique (minimally deemed suitable by an strength and conditioning expert, but nonetheless subject to individual variation and imperfections, thus reflecting real-world situations) have been explained as of the first revision.

Reviewer 3 Report

Thank you for taking the time to methodically work through my suggestions... excellent work and it has made my job far easier.

Author Response

We are pleased to read that you are satisfied with our work. Thanks again for taking the time to review it.